# OpenReview forum: "PaperBench: Evaluating AI’s Ability to Replicate AI Research"
_ICML.cc/2025/Conference — ICML 2025 poster_

### Official Review · Reviewer_HP2Q · 2025-02-16

**Overall Recommendation:** 4

**Summary:**

The authors have created a benchmark called PaperBench that tests model ability to reproduce modern machine learning research papers. The dataset consists of 18 papers (and a small dev set). To track granular progress on result reproduction, the authors create a hierarchical rubric for each paper. Judging outputs is done with an LLM, and the authors evaluate how well this performs compared to human judgements (pretty well). The authors evaluate 4 LLMs on the benchmark and find that their performance lags behind that of humans.

**Claims And Evidence:**

The authors are proposing a benchmark, not really testing claims. Claims regarding human and model performance seem reasonable.

**Essential References Not Discussed:**

I don't know of any

**Experimental Designs Or Analyses:**

No

**Methods And Evaluation Criteria:**

Yes

**Other Comments Or Suggestions:**

* One line 328 I think 5.3 is a typo. The link doesn't go anywhere.

**Other Strengths And Weaknesses:**

* Other strengths
  * The greatest strength of this paper is that it takes a first stab at an interesting and hard problem in a reasonable way
  * The scoring tree idea is intuitive and reasonable
  * Well-written
* Other weaknesses
  * I'm a bit concerned about the longevity/usefulness of the benchmark
    * 13/18 rubrics still under review
    * More repos could be added that wouldn't be in blacklist
    * Papers and code repos could end up in pre-training data (this identified by the authors)
  * Seems like counting "overnight"-type computer running for humans puts them at a relative disadvantage because models could let something run while also working on a different part of the project
  * In general it would be nice to see more qualitative results/discussion - e.g., When did models finish (or give up at) time-wise? Which types of nodes did models do better on (based on the classification in paper)?
  * The paper has some points I'm unclear on
    * Why does the human line ultimately go down in Figure 4?
    * Why were LLMs stopped at 12 hours? It seems like progress was still being made at that point?
    * Why can't LLMs just upload results to their repo? I see you only consider executed submission, but it seems like from the paper that this includes the non-executed submission.
  * It would be nice to see further discussion of some points
    * Why are scores on some papers higher than others (Figure 3)? Is it something about the papers? About the scoring tree?

**Questions For Authors:**

Using this as a tl;dr spot for my evaluation. Recommending "accept". See reasons for and against (including some questions) in "Other Strengths and Weaknesses". This benchmark has some substantial limitations (also discussed by the authors), but I think it provides a useful starting point for measuring LLM abilities on this hard and interesting problem.

**Relation To Broader Scientific Literature:**

The authors discuss this thoroughly in the Related Work section and I know of no connections they don't identify

**Theoretical Claims:**

No proofs in this paper

---

> ### Author Rebuttal · Authors · 2025-04-01
>
> Thank you for your insightful feedback! We address your comments below:
>
> 1. Concerns About Longevity/Usefulness of Benchmark
>
> Thank you for raising this important point! We are happy to announce that we now have author approval on 20 rubrics and so now the dataset and results are on a 20 paper dataset. We agree that there are contamination risks both as more potentially helpful resources become available online to agents that can browse the web (that aren’t included in the blacklists) and as papers and code repos get included in pre-training data. Building better monitors can help detect occurrences of the former, but won’t be that helpful for pre-training contamination. We hope that in future the PaperBench dataset can be expanded to include new papers that can help the benchmark stay relatively uncontaminated.
>
> 2 + 4a + 4b. Human vs. LLM Time Accounting ("Overnight computer runs")
>
> We agree with your critique. We think there are a few factors that mean directly comparing human vs AI agent time can be confusing: for example, AI agent time can be influenced by API inference speed which is variable. Since the version of the paper that was initially reviewed, we have significantly expanded our human baseline experiments in our new manuscript. The updated version shows considerably improved human replication scores (now up to 41.4% from 22.0% previously) via extending the baselines from 1 week to 4 weeks which we believe is a more realistic assumption for a human attempting the task. These new human baselines demonstrate that the PaperBench replications, while challenging, are indeed attainable for skilled humans and the plots suggest that scores would continue to improve if humans were given more time. We discuss this in a revised Section 5.4. In these results (on a 3 paper subset) we find that our tested model, o1, plateaus after around the first hour which justifies our decision to impose a timeout of 12 hours. On these new human baseline results the human line continues to go up and doesn’t show signs of plateauing.
>
> 3. We have now included a table in the appendix which stratifies scores based on requirement type. We observe that models perform poorly on Code Execution and Result Analysis requirement types, while scoring better at Code Development nodes.
>
> 4c. Why can't LLMs just upload results to their repo?
>
> Thank you for raising this important criticism! In light of this we have changed the way we grade Result Analysis nodes. Now all Result Analysis nodes are judged only by looking at any files created or modified
> in the reproduction step (as well as docs, reproduce.sh, and reproduce.log). By forcing the judge to only look at newly modified files this better guards against the judge being fooled by results that are included in the preliminary submission (but this could be hacked around by an adversarial AI agent that wants to fool the judge).
>
> 5a. Why are scores on some papers higher than others?
>
> Some paper scores are higher than others due to the inherent differences in difficulty between replicating different papers. By having papers of different difficulty in the dataset, we prevent the benchmark from saturating quickly and get a better estimate of agent capabilities since we’re tracking the agent’s performance on a greater variety of tasks. The main benchmark metric is average Replication Score which reports the mean Replication Score across each paper.

---

> > ### Comment · Reviewer_HP2Q · 2025-04-03
> >
> > Thanks for the response, and I'm happy to hear the paper has seen continued development.

---

### Official Review · Reviewer_s7ck · 2025-03-12

**Overall Recommendation:** 3

**Summary:**

This paper contributes PaperBench, a new benchmark that replicates the code implementations of top-tier AI conference research papers, including their code, and the papers' results from running the generated code, including their analysis. It uses 18 spotlight and Oral papers. It leverages LLMs to automate grading of the benchmark and proposes an LLM as judge and specialized dataset including addendums to evaluate this benchmark.

**Claims And Evidence:**

* "We open-source our code". Code is not open sourced, or available to reviewers for review. I encourage the authors to fulfill this claim to allow others to use this benchmark.
  * "PaperBench: a dataset of 18 machine learning papers, each accompanied by a human-curated and author-approved rubric". The contribution of this paper is a new benchmark PaperBench; however, given the code is not contributed with the paper, the rubrics are not given either in the paper therefore, this claim is untrue as it stands. I encourage the authors to release their code, or provide the full rubrics within the appendix of the paper to allow the Benchmark to be used and implemented by others.
* "JudgeEval: an auxiliary evaluation that evaluates how accurately a given judge grades replication attempts." No novelty, as using LLM as a judge has been done before (Zheng et al. 2023).
* "measuring AI agents’ abilities to conduct long-horizon tasks and engage in machine learning research and development.". This claim should be tightened, as it is too broad, as although PaperBench measures how well an LLM with agent scaffolding can replicate the codebase to replicate a paper, it is not actually doing ML research and development, as the paper is given to the LLM when replicating the codebase, which contains the method and any other research findings. If the authors truly wish to create a benchmark for ML research evaluation, this would be similar to AI Scientist (Lu et al. 2024).


References:

* Zheng, Lianmin, et al. "Judging llm-as-a-judge with mt-bench and chatbot arena." Advances in Neural Information Processing Systems 36 (2023): 46595-46623.
* Lu, Chris, et al. "The ai scientist: Towards fully automated open-ended scientific discovery." arXiv preprint arXiv:2408.06292 (2024).

**Essential References Not Discussed:**

Most relevant related work is not discussed ("Replicating a High-Impact Scientific Publication Using Systems of Large Language Models.", Bersenev et al. 2024), and clear delineation on novelty could be beneficial to readers.

Additional citations that help the reader would be citing existing work in the area of using an LLM as a judge, e.g., (Zheng et al. 2023), and differentiating and refining the core claim to do scientific research which this is not, as scientific research papers include (Lu et al. 2024).



* Bersenev, Dennis, Ayako Yachie-Kinoshita, and Sucheendra K. Palaniappan. "Replicating a High-Impact Scientific Publication Using Systems of Large Language Models." bioRxiv (2024): 2024-04.
* Zheng, Lianmin, et al. "Judging llm-as-a-judge with mt-bench and chatbot arena." Advances in Neural Information Processing Systems 36 (2023): 46595-46623.
* Lu, Chris, et al. "The ai scientist: Towards fully automated open-ended scientific discovery." arXiv preprint arXiv:2408.06292 (2024).

**Experimental Designs Or Analyses:**

The proposed Benchmark was benchmarked correctly to it's outlined implementation.

**Methods And Evaluation Criteria:**

* The methods and evaluation criteria make sense for a new benchmark contribution; however, the evaluation aspects of the benchmark and its construction can all be better motivated and improved.
  * Paper lacks motivation why the tree rubric structure was chosen over a simple checklist structure, was this used in other papers, perhaps? Also how are the weights determined in the rubrics, do they have a particular meaning, should the weights follow a specific structure? Why are the weighted nodes needed over uniform nodes?
  * Judge LLM Agent only reads top-k code files, where k=10. Why was 10 selected, can you motivate this choice through an ablation of different k's. Or are alternative approaches superior such as scanning through multiple files with that rubric until you find files that satisfy it? Why not avoid this, and use a larger context window size LLM? Such as LLMs that can process 2 million tokens.
  * How many random seeds, or independent runs were used? To calculate the error bars in the table; this was not clear from the paper.

**Other Comments Or Suggestions:**

* Why is the human replication rate so low? If so low, how can we expect LLM Agents to solve it?
* Can you inspect which websites it visited in the logs? How do you ensure no leakage by visiting blogs discussing the paper or even other code bases that can have partial or re-implementations of it?
* Assumption of un-restricted compute, seems difficult for others to use as a benchmark.
* Can any consistency be imposed across the rubrics?
* Can future versions also incorporate multi-modal LLMs to analyze plots too?
* What about using the benchmark with small context LLMs? We could benefit from a discussion on the suitability and requirements for an LLM to be used in the benchmark.
* It seems the scores should be normalized per paper, rather than absolute counting.
* Can you run a sanity check where you use your judge on the actual repo produced
 (plus any addendums) by the paper to check if the replication score is 100%? If not then the heuristics are invalid.
* "AISI’s Basic Agent scaffolding" is not described in the paper, or cited, however there is a link, paper would benefit from an explanation of this scaffold to be self contained.
* No guarantee or test that the addendum is fully correct; as it was created by when creating the rubrcics. Need a better sanity check to ensure that it is correct.
* Ablation providing the agent with the Judge addendum; does it improve the score? If so, does this indicate poor retrieving and searching for relevant papers that help with implementing the algorithm/paper that were accessible when the paper came out, that the original authors may have had access to those papers.
* Explain relative performance amongst LLMs, and any insights there, could be helpful for the reader.

**Other Strengths And Weaknesses:**

* Clarity: Paper is well written, and clear to follow.
* Originality: Paper replication benchmark is novel, however it lacks proper motivation on why certain elements of the benchmark were chosen or references to existing benchmarks constructed, see above comments.
* Significance: Having a benchmark to replicate papers is valuable for the community, however it could be made more valuable by also thinking how this approach could be made fully scalable, and if the rubrics and any addendums can be self generated for new papers, to improve on the problem that the benchmark contains only a few papers (18). Additionally, it would be helpful for the community if the authors could analyze the failures cases and provide insight into why the scores are so low, what tasks on average the LLM agent gets right, and what are the current open challenges that it struggles with? If agent scaffolding is the big problem as discussed in the paper, can the authors investigate ablations with different agent scaffolding techniques (Huang et al. 2024)?

References:

* Huang, Xu, et al. "Understanding the planning of LLM agents: A survey." arXiv preprint arXiv:2402.02716 (2024).

**Questions For Authors:**

1. Analysis into the failures? Why is it so low? Do these papers rely on specific external libraries that the LLM doesn’t know about, such as transformers library etc; a particular dated version of a dependency, numpy, Jax, Tensorflow or PyTorch?
2. Analysis of successes: which three different types do we see succeeding and failing, and do any insights here?
3. Can you artificially scale up the process to create these heuristics from papers? If so how?

**Relation To Broader Scientific Literature:**

Key proposed contribution of this paper is a new ML paper codebase replication benchmark. The idea of using LLMs to replicate results and code from a paper is not novel, see ("Replicating a High-Impact Scientific Publication Using Systems of Large Language Models.", Bersenev et al. 2024), however this specific ML related replication benchmark is novel. However the authors should cite this paper, and any others that already perform replication of papers.



References:

* Bersenev, Dennis, Ayako Yachie-Kinoshita, and Sucheendra K. Palaniappan. "Replicating a High-Impact Scientific Publication Using Systems of Large Language Models." bioRxiv (2024): 2024-04.

**Theoretical Claims:**

* Not applicable, no theoretical claims are made

---

> ### Author Rebuttal · Authors · 2025-04-01
>
> Thank you for taking the time to carefully review our paper! The rebuttal is subject to a character limit; we address what we saw as the highest-priority comments below:
>
> > Code is not open sourced, or available to reviewers for review.
>
> We will open source our codebase for the camera-ready release. The codebase can be found under the supplementary materials, which has been available during the review process.
>
> > No novelty, as using LLM as a judge has been done before
>
> JudgeEval is a necessary auxiliary evaluation in order for us to justify using an LLM-based judge for the grading of paper replication attempts. We’d like to stress that we’re not claiming the use of a judge is a novel idea.
>
> > This claim should be tightened, as it is too broad…
>
> We now use updated wording in the Abstract and Introduction which clearly states the intended scope of PaperBench: we claim that it is “a benchmark evaluating the ability of AI agents to _replicate_ state-of-the-art AI research”.
>
> > …the evaluation aspects of the benchmark and its construction can all be better motivated and improved.
>
> 2a. Why use a tree rubric structure, and how weights are determined:
>
> The tree structure allows us to measure partial progress towards a result in a paper (allowing us to determine whether agents are scoring highly on particular sub-trees) and hierarchically decompose the grading task into smaller (and therefore easier) tasks for the Judge.
>
> Rubric weights are carefully set by each rubric creator and are author-reviewed to reflect the relative importance of each sub-task.
>
> 2b. Judge LLM Agent only reads top-k code files, where k=10:
>
> Our judge achieved an F1 score of 0.83 on JudgeEval using $k=10$. Preliminary experiments showed larger values of $k$ yielded similar performance, while smaller values lowered the F1 score. Future work exploring alternative judge designs (e.g., longer-context or vision-capable models) would be valuable.
>
> 2c. Clarifying Number of Seeds and Runs ("How many random seeds?")
> In Section 5.2 of our new manuscript, we clarify that we performed evaluations with three independent runs for each evaluated agent to robustly estimate performance and error bars.
>
> > However the authors should cite this paper...
>
> Thanks for pointing us towards Bersenev et al.’s work; we will reference it in related work.
>
> > Significance: …if the rubrics and any addendums can be self generated for new paper…
>
> We’ve reached the character limit for this rebuttal, so please see our detailed response to Reviewer b2MF’s comment: “Not scalable: each paper requires…”
>
> > Why is the human replication rate so low? If so low, how can we expect LLM Agents to solve it?
>
> We’ve since improved the human baseline experiment, extending the best performers after week one to continue their attempt for a total of four weeks, a more realistic timeframe for humans. This led to an increase in the average replication score from 22.0% to 41.4%. In our updated Figure 3, we see a steady increase in the human baseline score over time, which we expect to continue given more time.
>
> > Assumption of un-restricted compute, seems difficult for others to use as a benchmark.
>
> We agree that running PaperBench with our setup is too resource intensive for many users. We therefore introduce a variant in our updated manuscript, PaperBench Code-Dev, where models are only evaluated on Code Development nodes. Users therefore don’t need a GPU or an expensive VM, which cuts costs substantially.
>
> > Can any consistency be imposed across the rubrics?
>
> We had multiple processes in place to ensure rubrics are created in a consistent manner. We adhered to team-wide guidelines for rubric creation. We also had a core team of reviewers who reviewed every rubric in the dataset, ensuring consistency. Additionally, the author of each paper reviewed the relevant rubric for correctness, often resulting in many iterations of refinement.
>
> > Can you run a sanity check where you use your judge on the actual repo produced…
>
> In JudgeEval, we manually graded original author repositories and found replication scores below 100%. As explained in the paper, this is expected since original codebases often have bugs, are incomplete, or lack the required reproduce.sh scripts.
>
> > No guarantee or test that the addendum is fully correct…
>
> Each addendum was co-developed with and reviewed by the original authors of each paper – often going through multiple rounds of iterations of refinement – which ensured they were high-quality and accurate.
>
> > Explain relative performance amongst LLMs…
>
> Thanks for the suggestion! We’ve updated the paper to briefly analyze why models perform differently from one another. Most models frequently finished early or faced a problem they couldn’t solve. All agents failed to verbalize how best to use the limited amount of time available to them.
>
> —
>
> Thank you for your valuable feedback! Please consider raising your rating if you feel that the paper has improved.

---

### Official Review · Reviewer_b2MF · 2025-03-12

**Overall Recommendation:** 2

**Summary:**

The paper introduces PaperBench, a benchmark for evaluating AI agents’ ability to replicate SOTA ML research. The dataset comprises 18 papers from ICML 2024. Each paper is accompanied by a manually curated rubric, which hierarchically decompose each replication task into smaller gradable subtasks. The an LLM-based judge to grade replication attempts. Experiments on proprietary LLMs show that the best-performing LLM -- Claude Sonnet 3.5 -- achieves a replication score of 14.1%, below the 22.0% achieved by human PhD participants on a five-paper subset. The study highlights the challenges AI faces in long-horizon ML research tasks and suggests that AI models can perform some research tasks but still fall short of human-level capabilities.

**Claims And Evidence:**

Overall, the claims made in this submission are convincingly supported by the presented evidence. However, the intermediate nature of the results (13 out of 18 rubrics under review by authors) somewhat limits the certainty of final conclusions. Additionally, the limited number of human annotators and AI model evaluations constrain the robustness and generalizability of the benchmark conclusions.

**Essential References Not Discussed:**

Line 47 mentions the UK AI Safety Institute’s Basic Agent scaffolding without an explicit citation, which should be included to clarify the source and structure of the agent scaffolding.

**Experimental Designs Or Analyses:**

Yes. Including comparing models under controlled conditions and comparing proprietary LLMs such as o1 and Claude 3.5 Sonnet.

**Methods And Evaluation Criteria:**

Yes

**Other Comments Or Suggestions:**

Line 47: Lack of citation - UK AISI’s Basic Agent scaffolding

**Other Strengths And Weaknesses:**

Strengths:
1. Propose a hierarchical rubrics that enable precise measurement of partial replication progress.
2. Transparent evaluation and clearly documented processes, including the introduction of JudgeEval.
3. Evaluate many state-of-the-art models on PaperBench, measuring AI agents’ abilities to conduct long-horizon tasks and engage in machine learning research and development.

Weaknesses:
1. Very limited annotations and data: a) only 18 papers, and b) only 8 submissions from PhD students to establish human baseline performance. This restricts the benchmark's representativeness and reliability.

2. Not scalable: each paper requires a manually created rubric and associated weights, raising concerns about scalability due to the intensive manual effort involved.

3. Very expensive for subsequent researchers to utilize the benchmark: grading a single submission costs 150 USD, and evaluating 18 submissions would cost $2,700 for a single model or method. This high cost is prohibitive for others aiming to benchmark their models, thus limiting the benchmark's broader impact.

4. Lack of experiments involving prominent open-source models such as DeepSeek-R1 or LLaMA, restricting the generalizability of the findings.

**Questions For Authors:**

1. You mention that 13 rubrics are currently still under review by paper authors, making your presented results an intermediate evaluation. Could you clarify how significantly you expect the final results to differ from the current intermediate evaluation?

**Relation To Broader Scientific Literature:**

The paper clearly situates its contribution within related benchmarks such as CORE-Bench, MLE-bench, MLAgentBench, and RE-Bench, emphasizing that PaperBench uniquely assesses AI’s ability to replicate current state-of-the-art research papers autonomously and from scratch.

**Theoretical Claims:**

The paper does not explicitly contain theoretical claims or proofs.

---

> ### Author Rebuttal · Authors · 2025-04-01
>
> Thank you for your constructive feedback! We address your comments below:
>
> > Line 47 mentions the UK AI Safety Institute’s Basic Agent scaffolding without an explicit citation, which should be included to clarify the source and structure of the agent scaffolding.
>
> Thank you for catching this! We’ve added a citation for UK AISI’s Basic Agent scaffolding and have explicitly referenced it throughout the paper.
>
> > Very limited annotations and data: a) only 18 papers, and b) only 8 submissions from PhD students to establish human baseline performance.This restricts the benchmark's representativeness and reliability.
>
> Thank you for the feedback! We’ve updated PaperBench to include 20 papers (up from 18), all of which have been finalized with author approvals. The rubric now has 8,316 individually gradable tasks (up from 6,125). We’d like to mention that we think the number of individually gradable tasks gives a more accurate picture of how diverse the tasks in PaperBench are, rather than the number of papers.
>
> We’ve also improved the human baseline experiment, extending the best performers after week one to continue their attempt for a total of four weeks, since we think this is a more reasonable timeframe for humans to replicate a paper. Considering the turnaround time and cost of running such a human baseline experiment, it was infeasible to conduct it with a larger sample size.
>
> > Not scalable: each paper requires a manually created rubric and associated weights, raising concerns about scalability due to the intensive manual effort involved.
>
> We agree that each rubric involves a significant amount of effort to create. We found that current models aren’t capable of end-to-end rubric creation (even after extensive prompt iteration) and, even when rubrics were generated end-to-end, they took a significant amount of time to review. Models were, however, useful in assisting humans in writing rubrics. We have released our rubric creation web application in the supplementary materials, which uses models to assist humans in the rubric creation process; we used this tool throughout our rubric-creation process. We’ve also added Appendix A.1 which outlines future directions of automated rubric creation, which may be possible in the near future.
>
> > Very expensive for subsequent researchers to utilize the benchmark: grading a single submission costs 150USD and evaluating 18 submissions would cost 2,700 for a single model or method.This high cost is prohibitive for others aiming to benchmark their models, thus limiting the benchmark's broader impact
>
> Thank you for raising this point! We’ve significantly reduced the cost of grading to approximately 66 USD per paper (down from 150 USD) by using a cheaper model (o3-mini-high) for the Judge. Importantly, using o3-mini-high achieves comparable scores to the previous model on JudgeEval. In general, we expect the cost of grading to fall sharply as cheaper and more capable models become available in the future, meaning this barrier to entry will be short-lived.
>
> We also release a lighter-weight variant, PaperBench Code-Dev (which supersedes PaperBench Lite and is even more accessible and affordable), reducing the grading cost further from 66 USD per paper to approximately 10 USD per paper when using our optimized o3-mini based judge. We describe this new lightweight variant in detail in Section 2.6, but in short: it’s a lighter-weight variant of PaperBench designed for situations where you only want to see if the agent can write plausibly-correct code, without the overhead of running it. Grading costs are significantly reduced, since the judge grades fewer nodes, and doesn’t need to place the reproduce.log file in its context (which can be lengthy).
>
> We’ve also created an experimental grading feature whereby we grade rubrics which are “pruned” to a given depth, potentially reducing the cost of grading by 10x or more. We’ve added a new section, Appendix H, which covers this approach in more detail, including experimental results on JudgeEval.
>
> > Lack of experiments involving prominent open-source models such as DeepSeek-R1 or LLaMA, restricting the generalizability of the findings.
>
> Thank you for the suggestion! We’ve updated the paper to include a leading open-source model, DeepSeek-R1. We find that DeepSeek-R1 performs better than some closed-source frontier models (e.g., GPT-4o) but worse than others (e.g., Claude Sonnet 3.5).
>
> > You mention that 13 rubrics are currently still under review by paper authors, making your presented results an intermediate evaluation. Could you clarify how significantly you expect the final results to differ from the current intermediate evaluation?
>
> We've now completed all author reviews of rubrics, have re-run our experiments and found that our results are broadly consistent with the preliminary results presented originally.
>
> —
>
> Thank you for your valuable feedback! Please consider raising your review score if you feel that the paper has improved.

---

> > ### Comment · Reviewer_b2MF · 2025-04-04
> >
> > Although I thank the authors for their response, unfortunately, my concerns about the extremely prohibitive evaluation cost, limited paper coverage, and scalability still remain. I would like to keep my current score.

---

> > > ### Author Response · Authors · 2025-04-07
> > >
> > > Thank you for your follow-up response. We respectfully disagree with the characterization of PaperBench as having an “extremely prohibitive” evaluation cost.
> > >
> > > We understand that PaperBench, like other agent-based benchmarks, has higher costs compared to traditional QA-style evaluations due to the longer task horizons and corresponding token usage required. However, these costs remain within the typical range for comparable agent benchmarks widely adopted in the research community. For instance, the creators of the Aide coding agent estimated evaluation costs of approximately 10k USD for SWE-bench Verified [1], and developers of the OpenHands agent reported approximately 6k USD for running the full SWE-bench [2].
> > >
> > > In contrast, the lightweight PaperBench Code-Dev variant we introduced can similarly be run within budgets in the low single-digit thousands of USD, making it comparably accessible for typical academic research groups. While we recognize these costs might still limit certain researchers, we believe they are reasonably accessible and not unusually prohibitive when contextualized within the broader agent benchmarking space.
> > >
> > > We kindly ask the reviewer to clarify if there is a specific reason PaperBench’s evaluation costs are viewed as significantly more prohibitive compared to benchmarks like SWE-bench, which have similar or higher associated costs.
> > >
> > > Regarding the concern about limited paper coverage, we have previously noted that the most meaningful measure of PaperBench’s scope is the total number of individually gradable tasks (8,316), rather than just the paper count itself. The reviewer originally stated that the size of our dataset “restricts the benchmark's representativeness and reliability”; we invite the reviewer to elaborate on why our dataset is limited in its representativeness and reliability, despite having a large number of individual tasks across a range of sub-tasks for each paper.
> > >
> > > We sincerely appreciate your constructive feedback.
> > >
> > > [1] https://news.ycombinator.com/item?id=42638605#:~:text=Hey%21%20One%20of%20the%20creators,of%20Aide%20here
> > >
> > > [2] https://github.com/All-Hands-AI/OpenHands/issues/1693#issuecomment-2105057205

---

### Decision · Program_Chairs · 2025-05-01

**Decision:**

Accept (poster)

**Comment:**

This paper presents a novel and timely benchmark aimed at evaluating the replicability of papers, which is a critical yet underexplored dimension of scientific progress. The work involves a thoughtful design of LLM-based judge scaffolding, evaluation rubrics, human comparisons, and several insightful findings. The methodology is well-structured and clearly articulated, and the paper is well-written and easy to follow.

The majority of reviewers gave positive ratings and acknowledged the significance and originality of the work. One reviewer raised a valid concern about the operational cost and scalability of the proposed framework. While this is important to consider for real-world deployment, AC believes that the overall strengths—namely, the relevance, clarity, and impact of the benchmark—outweigh this limitation.

Another reviewer, while positive overall, made helpful suggestions regarding the influence of different agent scaffolding strategies on outcomes. Additionally, they noted parallels with SWE-Bench's development process. These points offer valuable opportunities to strengthen the paper.

Overall, AC recommends weak acceptance, and strongly encourages the authors to incorporate the reviewers’ suggestions—particularly regarding agent scaffolding comparisons —in the final version to further enhance the clarity and contribution of the work. Besides, the authors are strongly encouraged to acknowledge and discuss the trade-offs in scalability and operational costs more explicitly in the final version.